# Applications of Hydrogel with Special Physical Properties in Bone and Cartilage Regeneration

**DOI:** 10.3390/ma14010235

**Published:** 2021-01-05

**Authors:** Hua Lin, Cuilan Yin, Anchun Mo, Guang Hong

**Affiliations:** 1State Key Laboratory of Oral Diseases, National Clinical Research Center for Oral Diseases, West China Hospital of Stomatology, Sichuan University, No. 14, Section 3, South People’s Road, Chengdu 610041, China; lin.hua.q6@dc.tohoku.ac.jp (H.L.); 2017324035034@stu.scu.edu.cn (C.Y.); 2Division of Advanced Prosthetic Dentistry, Graduate School of Dentistry, Tohoku University, 4-1 Seiryo-machi, Aoba-ku, Sendai 980-8575, Japan; 3Division for Globalization Initiative, Liaison Center for Innovative Dentistry, Graduate School of Dentistry, Tohoku University, Sendai 980-8575, Japan; hong.guang.d6@tohoku.ac.jp

**Keywords:** bone regeneration, mechanical property, bioactive, biocompatibility, biodegradability

## Abstract

Hydrogel is a polymer matrix containing a large amount of water. It is similar to extracellular matrix components. It comes into contact with blood, body fluids, and human tissues without affecting the metabolism of organisms. It can be applied to bone and cartilage tissues. This article introduces the high-strength polymer hydrogel and its modification methods to adapt to the field of bone and cartilage tissue engineering. From the perspective of the mechanical properties of hydrogels, the mechanical strength of hydrogels has experienced from the weak-strength traditional hydrogels to the high-strength hydrogels, then the injectable hydrogels were invented and realized the purpose of good fluidity before the use of hydrogels and high strength in the later period. In addition, specific methods to give special physical properties to the hydrogel used in the field of bone and cartilage tissue engineering will also be discussed, such as 3D printing, integrated repair of bone and cartilage tissue, bone vascularization, and osteogenesis hydrogels that regulate cell growth, antibacterial properties, and repeatable viscosity in humid environments. Finally, we explain the main reasons and contradictions in current applications, look forward to the research prospects in the field of bone and cartilage tissue engineering, and emphasize the importance of conducting research in this field to promote medical progress.

## 1. Introduction

Hydrogel is a kind of polymer network system with water as dispersion medium and a lot of water in the matrix. It is insoluble in water due to its special cross-linked structure [1]. At the same time, hydrogels are very similar in physical and chemical properties to most human tissues and consist of polysaccharides and proteins [2,3]. Hydrogels usually not only do not affect the body’s metabolic processes, but also allow metabolites to pass through them to be excreted from the body [4]. They play an important role in regulating cell functions, including cell migration and binding to cell membrane receptors to transmit signals [5]. In addition, hydrogels absorb water and reduce friction and mechanical effects on surrounding tissues [6]. Therefore, hydrogels with controllable degradability, biocompatibility, and good mechanical properties have a broad application prospect [7]. The polymer system of hydrogels can be used for cell transplantation and differentiation, endogenous regeneration, and biological repair, wound healing, and continuous drug delivery providing a good matrix [8], and its three-dimensional network system can mimic the microstructure of the original extracellular matrix and provide living ecological conditions for cell survival [9]. In general, the mechanical properties of hydrogels are enhanced with the increase of polymer concentration, which indicates that the biocompatibility and degradation of hydrogels are low [10]. This contradiction limits the application of hydrogels in biomedicine. As we all know, the brittleness of hydrogels mainly comes from irregular structures. Designing a series of high-strength polymer hydrogels with different structures, such as structures with regular or special networks, has become a solution to this problem [11]. With the deepening of research, hydrogels with good physical properties have been developed, which have obtained new applications in the field of biomedicine and promoted the development of medicine (As shown in Table 1) [12]. This article describes the morphology and mechanical properties of hydrogels used in bone and cartilage engineering. The application of hydrogels with special physical properties prepared by different methods in osteochondral tissue engineering are discussed.

With the rapid development of hydrogels in the field of bone and cartilage tissue engineering, the objective of this review is to cover the current emerging design development of hydrogels for repairing bone and cartilage defects. This report is not comprehensive, but it considers the outstanding representatives of this field in recent years, especially focusing on the water gel changes in the design and processing performance, what makes it have strong mechanical properties, and the fact that it can adapt to the bone and cartilage tissue engineering high mechanical performance requirements. We also explore application categories listed in the hydrogel material characteristics.

## 2. Materials and Methods

### 2.1. Search Strategy

An electronic search was performed by two reviewers (H.L. and C.Y.) from 1991 to May 2020, using Medline (PubMed), ScienceDirect for English articles. The search strategy applied was a combination of medical subject headings (MeSH) terms and free text words, including the following keywords: hydrogel, bone, cartilage tissue engineering, mechanical property, traditional network, double network hydrogels, chemical/ionic double crosslinking, injectable hydrogel, shear-thinning, bioactive, biocompatibility, biodegradability, 3D printing, integrated hydrogel, vascularization, antibacterial, viscous, and viscoelastic hydrogels.

The option of “related articles” was also used. Review articles, as well as references from different studies, were also used to identify relevant articles (Figure 1a). A graph representing the publication of hydrogel applicability shown with years can be found in Figure 1b. It is an overview in a glance to the increase in demand of hydrogel. In the end, a total of 95 articles were included in the study.

### 2.2. Selection of Studies 

The review process was divided into two stages. In the first stage, the searched titles and abstracts were first screened for relevance, and then the full text of the relevant abstracts was evaluated. Any disagreements were resolved by combining the opinions of the final inter-reviewer. In addition, the selected journals were searched manually, and references of selected studies were searched. In the first step of the review process, the following exclusion and inclusion criteria were used to screen the obtained articles:

#### 2.2.1. Inclusion Criteria

An origin of hydrogel.Cell, bone, and cartilage subjects.The content of the article should match the application of hydrogel to bone and cartilage tissue engineering.Review articles that help identify articles related to bone and cartilage regeneration.

#### 2.2.2. Exclusion Criteria

Only human studies.Finite element analysis studies.Case reports or case studies.Studies not providing detailed information on the properties of hydrogels and their applications in bone or cartilage engineering.

In the second stage of the review process, two reviewers independently screened the selected full texts based on the following exclusion criteria:When multiple similar articles were found in the same research, the latest research was included.When data on the success rate of hydrogels in animal experiments cannot be extracted from the study.Clinical studies in which the hydrogel was polymerized in vivo.Studies that only considered the cumulative success rate of hydrogel implantation or only considered biological complications without considering the actual clinical implementation requirements were excluded.

## 3. Results

### 3.1. Polymeric Hydrogels and Their Mechanical Properties

#### 3.1.1. Traditional Network Hydrogel

Bone healing is a complex physiological process triggered by the early regulation of inflammatory immunity, including angiogenesis [13], osteogenic differentiation, and biomineralization [14]. Hong et al. [15] invented an elastic hydrogel that can be used for three-dimensional bioprinting. The hydrogel is based on a single-network polymer, has high flexibility and elasticity, and can be gelled under blue light irradiation. The previously existing double-network high-elasticity hydrogel has a single network structure and a single gel formation excitation factor (blue, visible light), thus greatly simplifying the process of three-dimensional bioprinting. Fan et al. [16] have developed an injectable extracellular matrix (PEM) hydrogel that dynamically integrates multiple biological functions and acts at different stages of the fracture healing process. Compared with type I collagen hydrogel, PEM hydrogel promotes the transformation of macrophages from M1 to M2. PEM hydrogel promotes blood vessel migration, the development of large blood vessels, and the formation of functional blood vessels. PEM hydrogel can promote the formation of mature bone in bone defect better than type I collagen hydrogel. However, the traditional methods of strengthening hydrogels mentioned above cannot meet the load-bearing requirements of human bone and cartilage tissues. In recent years, many methods have been developed to enhance gel strength and further improve gel stability. To solve the problem of poor mechanical properties of hydrogels, researchers proposed a variety of methods to improve the gel strength, such as slip-ring hydrogels [17], nanocomposite hydrogels [18], macromolecular microsphere enhanced hydrogels [19], four-arm polyethylene glycol hydrogels [20], double-network hydrogels [21], etc. However, the traditional strength enhancement methods mentioned above cannot meet the load-bearing requirements of human tissues. In recent years, many methods have been developed to enhance gel strength, such as new double-network hydrogels and chemical/ionic double-crosslinked hydrogels.

#### 3.1.2. Double Network Hydrogels with High Strength

Compared with traditional single network hydrogels, double network hydrogels have better comprehensive mechanical properties. Gong et al. [22] of Hokkaido University in Japan combined the mineralized hydroxyapatite nanospheres (200–600 nm in diameter) with a high-strength double-network gel (Figure 2). This structure enhances the spontaneous osteogenesis ability of the gel surface, strengthens the combination with bone, and can be used as cartilage and artificial ligament [23]. Gu et al. [24] invented a hydrogel containing sodium alginate (Alg), polyacrylic acid (PAA), and demineralized bone matrix (DBM) to form an interpenetrating network. DBM is an allograft bone that removes inorganic minerals. It has better biological activity during the demineralization process, can regulate differentiation, and plays an important role in bone regeneration. The resulting hydrogel with covalently crosslinked polyethylene glycol diacrylate (PEGDA) and ionically crosslinked alginate has high strength and promotes cell proliferation. Gong et al. [25] also designed a double-network high-strength hydrogel based on amphiphilic triblock copolymer, whose design concept was to synthesize a triblock copolymer gel with hydrophobic chain at both ends and hydrophilic chain in the middle. Zhao et al. [26] synthesized gel with high strength, high toughness, and high tensile ratio and self-adhesion properties by using unstructured elastin. Elastin-based gels with tetra-covalent crosslinking points were first synthesized, then added with lysine groups that can complexate with zinc ions, and finally the gels were immersed in zinc sulfate solution. These two effects give the non-structural protein-based hydrogels high strength and toughness properties. However, the formation of this gel requires a high ion concentration, and although the yield of non-structural proteins is high and the gel has good biocompatibility, it is not suitable to be used as a biological material in the field of bioengineering.

#### 3.1.3. Chemical/Ionic Double Crosslinking High Strength Hydrogels

Suo et al. [27,28] prepared the physical/chemical double-crosslinked double-network hydrogels through a simple one-pot method. The hydrogels with high tensile ratio and toughness were prepared by photo-initiated polymerization, with alginate and acrylamide as monomers and a small amount of methylene bisacrylamide (MBAA) as crosslinking agent mixed with calcium ions. When bearing the external force, the ionic bond first breaks, which distributes most of the energy, so that the tensile green reaches more than 20 times and the tearing energy is close to 9000 J/m^2^, even in the case of a gap, it can still achieve 17 times of the tensile, based on the increase of the ion/covalently crosslinked hydrogel strength and the resilience. Zhou et al. [29] use acrylic instead of sodium alginate, macromolecular monomer with ferric iron ion to implement this kind of negative ion crosslinking effect, and further by immersion process, make the trivalent crosslinking of iron ion reassembled into a more neat and orderly structure, and give some 6 megapascal (MPa) tensile strength, with tensile rate more than 7 times.

#### 3.1.4. Injectable Hydrogel

The high strength hydrogels above meet the mechanical requirements of bone and cartilage tissue engineering on a macroscopic level, but neither biological nor synthetic systems are static, they are dynamic. At the cellular level, numerous studies over the past two decades have demonstrated that the elasticity or stiffness of the extracellular matrix (ECM) affects basic cellular processes, including diffusion, growth, proliferation, migration, differentiation, and organic-like formation [30].

At the level of cell mechanics, the self-healing bio-ink hydrogel was invented based on shear dilution, which simulates the complex physiological microenvironment. For extrusion-based bioprinting, how to provide high-reliability and high-resolution printing structures while ensuring that cells are not subjected to shearing forces during the printing process has attracted more attention. Nasim et al. [31] proposed that due to the interaction between glycosaminoglycan nanoparticles (GAGNPs) negative charge groups and laponite (LA) edges, nanocomposite hydrogels can be formed rapidly within a few seconds. The shear thinning behavior of hydrogel protects encapsulated cells from the influence of erosive shear stress during the process of bio-printing. Bio-inks can be printed directly into durable, independent structures with a high aspect ratio.

Hydrogels can be injected because of their shear thinning properties [32]. However, the poor mechanical properties and biological activity of some injectable hydrogels greatly hinder their clinical application. Liu et al. [33] developed injectable nanocomposite hydrogels for bone regeneration by growing calcium phosphate (CaP) nanoparticles and in situ growth of calcium phosphate NPs (ICPNs) during radical polymerization of dimethylaminoethyl methacrylate (DMAEMA) and 2-hydroxyethyl methacrylate (HEMA) matrix (PDH). The results showed that the best nanocomposite hydrogels obtained showed appropriate injection time and enhanced tensile strength of 321.1 kPa and fracture capacity of 29.0 kJ/m^2^. Compared with the hydrogels prepared by mixing non-in-situ prefabricated CaP NPs (ECPNs), they significantly promoted cell adhesion and upregulation of bone (Figure 3). Ingavle et al. [34] found that mesenchymal stromal cells (MSC) are captured in a composite hydrogel based on two naturally derived polymers (alginate and hyaluronate) containing biomineralized polymer microspheres. The presentation of the adhesion tripeptide arginine-glycine-aspartic acid (RGD) from the two polymers induced greater osteogenic differentiation of sheep MSCs in vitro.

### 3.2. Research Focus

#### 3.2.1. Bioactive

##### Ion-Containing Composite Hydrogel

Magnesium is the fourth most abundant metal element in the human body, half of which are found in bone tissue [35]. In natural bone, magnesium cations can enter the hydroxyapatite (HA) lattice and directly affect bone density and bone mechanics performance [36]. In recent years, researchers have conducted in-depth studies on the biological effects of magnesium ions and found that a certain concentration of magnesium ions can improve the proliferation and osteogenic differentiation of bone marrow mesenchymal stem cells [37]. However, excessively high concentrations of magnesium ions can also cause significant cytotoxicity and physiological toxicity [38]. In response to this problem, Cai et al. [39] prepared a series of PLGA\MgO\MgCO_3_ (PMg) composites using poly (lactic acid-glycolic acid) (PLGA), magnesium oxide (MgO), and magnesium carbonate (MgCO_3_) as raw materials. The release of Mg^2+^ can be controlled by the microsphere (Figure 4). The results of in vitro experiments show that PMg microspheres are non-cytotoxic and PMg-III, which has precisely regulated the release of magnesium ions, significantly promotes the migration, attachment, proliferation, and osteogenic differentiation of bone marrow mesenchymal stem cells. Generally, the rapid reaction between the carboxyl groups of alginate and calcium cations allows the formation of ionically crosslinked hydrogels. However, due to the rapid reaction of calcium ions, calcium cations cannot be directly introduced into the alginate solution [40,41]. Recently, a new method was developed to make homogeneous alginate hydrogels, which is to gradually release calcium cations from CaCO_3_ by reacting with the protons produced by the hydrolysis of gluconate d-lactone (GDL) [42].

##### In-Situ Mineralized Hydrogel of HAp Nanocrystals

Through a simple one-step micellar copolymerization of acrylamide and urethane dextran methacrylate (Dex-U), and then in-situ mineralization of HAp nanocrystals, a strong, tough, and osteoconductive material was developed for hydrogels. The introduction of hydrophobic micelle copolymerization and rigid crosslinked Dex-U phase gives the soft polyacrylamide (PAAM) network higher strength and toughness. In-situ mineralized HAp nanocrystals further improved the mechanical properties of the hydrogel [43] and promoted the later osteogenic differentiation of cells [44]. Mechanical tests and in vitro and in vivo evaluations confirmed that this HAp-mineralized PAAM/Dex-U hydrogel (HAp-PADH) has excellent mechanical properties and excellent osseointegration, so it can provide hope for bone repair and regeneration [45]. Hu et al. [46] studied the effects of HAp and gelatin microspheres (GMs) on the properties of gel scaffolds, including pH, microscopic morphology, gel time, mechanical properties, swelling rate, degradation behavior, and drug release.

##### Composite Hydrogel Encapsulated with Cells

By immersing the prepared hydrogel in a CaCl_2_/NaH_2_PO_4_ solution, calcium phosphate (CaP) nanoparticles can be grown in situ in the gel network [47]. This solution not only improves the mechanical properties of the hydrogel, but also effectively enhances its osseointegration and osteoconductivity [48]. However, this process requires the hydrogel to be immersed in a salt solution with a higher pH and osmotic pressure, so it is not suitable for hydrogels encapsulated with cells [49]. Bisphosphonates (BP) have excellent binding power with multivalent cations, inorganic nanoparticles, and mineralized tissues such as bone [50]. As the second most common divalent cation in cells, magnesium ions have multiple functions such as regulating enzyme activity and promoting cell adhesion and differentiation [51]. Bian et al. [52] prepared an injectable nanocomposite hydrogel containing cells through the interaction of BP and Mg, and through the secondary crosslinking for a specific time, the crosslinking density was further regulated to achieve the purpose of regulating the differentiation of stem cells (Figure 5).

#### 3.2.2. Biocompatibility

Biocompatibility refers to the ability of a material to cause an appropriate host response and material response in a specific practical application [53]. Ideal biological bone graft materials require good biocompatibility. Due to the intelligent type of hydrogel and its excellent biocompatibility, it is similar to the extracellular matrix in nature [51,54]. Studies have pointed out that stem cells are sensitive to the hardness and structure of the culture medium. Because of the special hydrophilicity, remarkable swelling behavior, and dual responsiveness of hydrogels [55], compared with other synthetic biological materials, it is closer to living tissues, and its friction and mechanical action on surrounding tissues after absorbing water is significantly reduced. It has good biological properties and is widely used in tissue engineering and organ construction [56]. Hyaluronic acid is an acidic mucopolysaccharide, which was first isolated from the vitreous body of a bull’s eye in 1934 by Meyer. Because of its unique molecular structure and physical and chemical properties, it shows a variety of important physiological functions: lens implantation [57], and anti-glaucoma surgery; it can also be used to treat arthritis and accelerate wound healing [58]. Researchers found that hyaluronic acid-based hydrogels have better biocompatibility and can improve cell viability [59]. Some scholars inoculated hepatocytes into galactosylated temperature-sensitive hydrogel, and found that the hydrogel can promote cell adhesion, proliferation, and function expression, namely the synthesis of specific albumin and urea, while the cell damage is small, and the ratio of vitality to glycation. A positive correlation can also reduce the inflammatory response of cells [60]. Studies [61] found that the composite hydrogel prepared by adding chitosan (CS) hydrogel to polyvinyl alcohol (PVA) hydrogel is more hydrophilic, and CS can improve the biocompatibility of the composite hydrogel. The collagen-polyvinyl alcohol (PVA) prepared by the repeated freezing and thawing method has good degradation performance, and the cytotoxicity of the biocompatibility test is 0 to 1. Jayakumar et al. [62] studied the role of dimethylallyl glycine (DMOG) in promoting cell migration and inducing the formation of capillary-like structures in human umbilical vein endothelial cells (HUVEC). By inducing osteogenic differentiation in rat adipose-derived mesenchymal stem cells (rADSCs), the effect of magnesite was studied, then the expression of osteogenic protein was analyzed using immunocyto chemical staining.

#### 3.2.3. Biodegradability

In polymer science and tissue engineering, the design of biodegradable polymers is becoming more and more important [63]. This is mainly for two reasons: polymers that can be naturally degraded into harmless products in the body can be used in biological devices. Biodegradability is critical for materials used in drug delivery devices or temporary structures in the body [64]. For these applications, the body’s ability to naturally degrade these materials is used as part of the application or part of the subsequent process of application. The key to the development of this material is to keep the bone and cartilage tissue regeneration and biodegradation synchronized [65]. Some other materials currently used are not effective, mainly due to the rapid degradation. The hydrogel Zheng et al. developed has strong controllability and it is achieved by adjusting the ratio of polyvinyl alcohol to fibrinogen. When the proportion of polyvinyl alcohol is high, it can slow down the biodegradation process and give the cells enough time to grow and heal [4,66]. In the initial stage of use, degradable polymer materials can provide the necessary mechanical support and guarantee physiological functions for the defect tissue [67]. As time goes by, in some cases, the role of the material is no longer needed, and the tissue begins to gradually repair at this time. The material gradually degrades due to its unique properties, the final tissue defect can be completely repaired, and the implanted polymer material can be degraded or absorbed and discharged from the body without unnecessary side effects or complications due to the accumulation of the material in the body [68]. Some scholars have found that methoxy polyethylene glycol polylactic acid (MPEG-PLA) hydrogel has good biocompatibility, and the use of materials will not cause the body to produce corresponding side effects and complications. Due to its special physical and chemical properties during the recovery process after surgery, it played a role in isolating the peritoneum and abdominal organs to prevent adhesion [69].

### 3.3. Applications

Injectable hydrogels must be cell-compatible and subject to restrictions such as the need to produce specific structures within a narrow temperature range and a limited mechanical shear force, and 3D printing enables control of these conditions. So far, however, due to the defects in mechanical properties of hydrogels, whether in wound dressing, tissue engineering or cell fixation, it has not been possible to obtain hydrogels with satisfactory mechanical properties and meet various requirements. The microscopic pore structure of hydrogel scaffolds is an important physical property, which not only affects the cell adhesion, growth, and the transport of oxygen, nutrients, and wastes during the growth process, but also the growth of host cells and blood vessels as well as the formation of new tissues. The hardness of hydrogels also has a great influence on bone tissue regeneration. When MSCs are cultured in hydrogels close to the hardness of nerve tissue ((0.1–1 kPa), muscle tissue (8–17 kPa), and bone tissue (25–40 kPa)), they show a high proliferation rate and differentiation ability, and can be differentiated in a specific direction.

#### 3.3.1. 3D Printing Hydrogel

Researchers used a bio-scaffolder printer to extrude ink through a 250 μm diameter nozzle to form a mesh stent with connecting parts, then put the stent into a crosslinking furnace to connect the particles, and then put it in deionized water to remove impurities [70]. This study [71] used 3D printing to inject gel and clay particles. The material, which is naturally similar to human bone, is used to treat different degrees of fractures. The basis of the material is the hydrogen bond, which shows highly flexible mechanical properties through ultraviolet reaction and attachable molecules (Figure 6). Dai et al. [72], combined with genome-wide association research and replication, the study found the genetic connection between growth differentiation factor 5 (GDF5) and developmental dysplasia of hip (DDH), and further used 3D bioprinting to make a scaffold loaded with GDF5 and nanoparticle cells, and used it as cartilage repair 3D bioprinting. To sum up, bio-inks for bone repair applications should have excellent printability, mechanical strength, and good biological activity. Zhao et al. [73] developed a new photosensitive nanocomposite bio-ink consisting of a triblock poly (lactide-propanediol—lactide) dimethacrylate (PmLnDMA), and a HEMA-functionalized nano-hydroxyapatite (nHAMA). With mechanical strength, biocompatibility, and bioactivity, it can be used for 3D printing bone tissue repair.

#### 3.3.2. Integrated Hydrogel for Simultaneous Repair of Cartilage and Bone

The prevalence of articular cartilage damage is high and it is difficult to self-repair after injury, and the existing articular cartilage repair materials cannot fully meet the clinical needs [74]. Because of their similarity to the extracellular matrix, wide adjustable range of water content, and excellent lubricity, hydrogels have potential application prospects as an articular cartilage repair material [75]. However, the osteochondral regeneration scaffold constructed by the prior art lacks the interface structure between cartilage and bone, and this interface layer plays important physiological functions such as buffering stress and maintaining the microenvironment [76]. Liu et al. [77] invented a high-strength hydrogel ink with enhanced hydrogen bond. The ink is based on acryloylglycinamide (PNAGA) copolymer supramolecular polymer hydrogel. PNAGA copolymer hydrogel has a lower melting temperature and better fluidity than its homopolymer hydrogel. The biohybrid gradient hydrogel scaffold still maintains a stable pore structure and good mechanical strength after being soaked in phosphate buffer (PBS) for a long time. Experimental results show that β-tricalcium phosphate (β-TCP) not only promotes the strength of the scaffold, but also effectively improves cell adhesion and proliferation, and differentiation towards the direction of osteogenesis endows the scaffold with good osteoinductive ability and forms a strong active bond with the host bone (Figure 7). The upper layer of the scaffold with higher water content can improve the lubricating properties of cartilage replacement materials, and the controllable load of transforming growth factor-β1 (TGF-β1) effectively promotes the differentiation of stem cells into chondrocytes. In vivo experiments show that the hybrid gradient hydrogel scaffold can simultaneously promote cartilage and subchondral bone regeneration.

#### 3.3.3. Hydrogels for Regulating Bone Vascularization and Bone Development

Bone is a highly vascularized tissue with a special and complex structure. Long bones are composed of a peripheral cortical shell, which contains a network of channels for blood vessel penetration and a highly vascularized bone marrow space inside [78]. Some researchers [79] have found a central 3D hydrogel construct that simulates bone, a gelatin methacrylate (GelMA) ring that contains peripheral octacalcium phosphate (OCP) to simulate the cortical shell, and a HUVEC spheroid that simulates bone marrow space. Designed as a GelMA ring (Figure 8), GelMA regulates the degree of formation of capillary-like structures derived from HUVEC spheroids. In conclusion, this type of cell-loaded hydrogel-based bone construct with a biomimetic double ring structure could be used in bone tissue engineering. Liao et al. [80] used biphasic calcium phosphate ceramic particles as mineralized bone to integrate hyaluronic acid-g-chitosan-g-poly (N-substituted butyramide) (HA-CPN) temperature-sensitive hydrogels. The complex can capture rabbit feces-derived stem cells and has osteoinductive ability, which can be transformed into new bone formation at the defect site after implantation of the implant into the rabbit skull defect.

#### 3.3.4. Antibacterial Compound Hydrogel

Bone regeneration and bone repair are a long-term process, in which the chronic infection problem has not been effectively solved [81]. In addition, growth factors can promote bone healing, but they are easily degraded in the body [82]. Therefore, simultaneously solving the problem of antibacterial and promoting bone healing is of great significance in orthopedic surgery [83,84]. Du et al. [85] designed a new peptide-like vesicle with excellent antibacterial properties, which kills bacteria and releases growth factors to promote bone healing. The researchers first designed a new type of peptide-like alternating polymer with excellent antibacterial properties and low cytotoxicity. The polymer can directly form vesicles in water and at the same time encapsulate the growth factor bone morphogenetic protein-2 (BMP-2). After that, the vesicles can attack and kill bacteria, release the growth factor BMP-2, and play a role in promoting bone regeneration. This article is a use of supramolecular hydrogel to encapsulate biologically active factors so that they can be continuously released in the bone defect area to promote the rapid regeneration of periodontal bone tissue. A well-studied hydrogel NapFFY was co-assembled with stromal cell-derived faceor-1 (SDF-1) and BMP-2 to prepare supramolecular hydrogels (SDF-1/BMP-2/NapFFY). In order to be able to load the above two growth factors and deliver them to the bone defect site, the author synthesized a supramolecular hydrogel using the non-covalent crosslinking self-assembly technology between small molecules, which has better biocompatibility and degradability than synthetic polymer hydrogels. Therefore, in this study, biocompatible supramolecular hydrogels were used to encapsulate SDF-1 and BMP-2. These were slowly released at the bone defect and activated in-situ regeneration and remodeling of periodontal bone tissue [86].

#### 3.3.5. Underwater Adhesive Hydrogel

When water molecules enter the bonding interface and form a water film, the direct contact between the adhesive and the substrate is severely restricted [87], resulting in a decrease in the surface energy of the adhesive, which in turn leads to a gap between the adhesive and the substrate. The bonding strength is significantly reduced or cannot be bonded at all [88,89]. The polyacrylamide hydrogel used in this work is a type of polymer material with a porous network structure [90]. A large number of crystals produced by the hydration of cement minerals haveled to the network structure and mechanical properties of the hydrogel composite material being obtained [91,92]. Once firmly bonded, the tight coupling between the nanocrystals in the hydrogel composite and the matrix is an important source of tight bonding between the adhesive and the matrix and high bonding strength [93]. Cui et al. [94] combines macroscopic surface modification with hydrogels with dynamic bonds to regulate the arrangement of supramolecular functional groups in the dynamic hydrogels on the surface of the hydrogels, so that the hydrogels have an effect on biological tissues. The quantified adhesion ability can be applied to wound dressings and adhesives for tissue healing (Figure 9). At the same time, the hydrogel preparation method is simple and easy to operate during the preparation process, and has long-term stability, which greatly expands the application of hydrogels in underwater or humid environments, especially in the potential of body fluids or blood environments [95]. Zhang et al. [96] summarized the technical difficulties in the underwater adhesion of hydrogel materials, focused on the relevant parameters that characterize the underwater mechanical properties of hydrogels, and established the experimental design and evaluation methods for hydrogel materials based on this, and summarized the unique properties, synthesis method, and preparation process of the biomimetic viscous hydrogel, and proposed new insights into the underwater adhesion mechanism.

## 4. Discussion

The purpose of the current systematic literature review was to evaluate the uses and prospects of hydrogels with special physical properties in bone and cartilage regeneration and to determine the types of hydrogels recommended for clinical application.

Hydrogels can be defined by swelling in the water and maintaining large amounts of water without being dissolved; it is a three-dimensional crosslinked polymer network structure, it can not only have strength in shape and softness to allow small molecules through, because some similar features and human body tissue hydrogels are widely used in bone and cartilage tissue engineering. However, due to the dispersion of a large number of aqueous media in the gel system and the heterogeneity of the network, the mechanical properties of the gel are poor, which greatly limits the application of gel. Many researchers have proposed innovative systems to enhance the strength of gels, which have greatly improved the tensile and compression properties of gels, such as tear energy, Young’s modulus, elongation, and so on. In recent years, many methods have been developed to enhance gel strength, such as new double-network hydrogels and chemical/ionic double-crosslinked hydrogels.

In practical clinical use, it is hoped that the hydrogel will have a low strength before being implanted into the bone or cartilage defect, so that it will have good fluidity. After being implanted into the defect, the hydrogel will be in a higher strength state. Injectable stimulation-responsive hydrogels provide an important platform for this goal. Injectable hydrogels encapsulate, manipulate, and transfer their cells and drugs to surrounding tissues in a minimally invasive manner. Although injectable hydrogels are easy to produce and have low cytotoxicity, generally they have slow response times and low stability. Over the past few years, many studies have focused on the synthesis of new injectable hydrogels for repairing cartilage and bone tissue. The main challenges are the injectable hydrogel bioactive scaffold required for cartilage and bone tissue engineering, full biocompatibility, biodegradability and stability, excellent mechanical properties for culturing three-dimensional cells, and how to design the transport of nutrients and growth factors. To address this challenge, in the first place, new injectable hydrogels can be developed using bioactive materials. Second, more advanced manufacturing methods need to be developed, primarily to improve mechanical properties and physiological stability and reduce cytotoxicity.

The incompatibility of some hydrogels may also affect the application of hydrogels in bone and cartilage tissue engineering. For example, polymethacrylate hydrogels, which are generally non-toxic in their own right, cause inflammation and tissue reactions mainly due to soluble substances in the polymer, which generally include residual raw monomers, low molecular weight polymers, catalysts, and other additives. Other hydrogels, which polymerize with ultraviolet light, can also cause cell death if the amount of photoinitiator is too high.

Previous studies on the mechanical properties of hydrogels mainly focused on the macroscopic level to ensure the mechanical strength of hydrogels, to ensure that hydrogels can withstand the compression force from bone and cartilage tissue. In recent years, more and more studies have focused on the mechanical effects of hydrogels at the cellular level. Viscoelastic hydrogels have been used for three-dimensional cell culture. How to precisely regulate the viscoelasticity of hydrogels may have a great influence on the physiological activities of cells. Hydrogels have several mechanics that play a key role in cellular activity, including cell proliferation and stem cell differentiation. These early results suggest that time-varying mechanics may be a unique and key parameter for regulating cell biology. In addition, depending on the viscoelastic mechanism of hydrogels, the importance of viscoelasticity and viscoplasticity affecting cellular behavior remains to be elucidated.

In order to construct complex three-dimensional physiological microenvironments, bio-inks based on shear thinning and self-healing hydrogels have been developed. For extruding-based bio-printing, it is a challenge to support high-reliability and high-resolution printing structures while ensuring that the cells are not subjected to shear forces during printing.

Although in recent years much correlation research has been carried out for high strength hydrogel, most studies focused on the simple increase gel strength; increasing polymer concentration can improve the mechanical strength, but result in poor biocompatibility and low degradability. This is the principal contradiction in the mechanical properties of the hydrogel in the field of biomedical applications. In previous studies, the introduction of non-chemical bonds has become an effective method to improve the strength of hydrogels, but the synergistic effect between chemical bonds and non-chemical bonds needs to be further studied. How to guarantee the mechanical strength; to further improve the biocompatibility of the hydrogel; how to maintain the stability of the hydrogels in the process of permanent replacement; how to implement gel in the temporary support process controllable degradation; how to better simulate the human body physiological environment; and realize the structure and function of bone and cartilage regeneration, while ensuring good biocompatibility and biodegradability, under the premise of development have special physical properties of water gel, is yet to be explored.

However, as a biomedical material, it only has high strength and activity, and its biological safety and biodegradability are far from meeting the actual requirements. Therefore, its biological function is an important index for the evaluation of biomedical materials. Many researchers have tried their hand at related fields. For example, 3D printing could be used to create defects specific to bone and cartilage to speed up tissue healing. Simultaneous defect of bone and cartilage is a common disease in clinical practice. The design of a hydrogel that can repair bone and cartilage defects at the same time is a research direction. Artificial grafts of bone and cartilage are often difficult to vascularize after implantation, so hydrogels that guide blood vessels could be a solution. At the same time, bone and cartilage are easy to be complicated with bacterial infection during wound healing, so it may have a good clinical application prospect to endow hydrogel with antibacterial properties. Although adhesion hydrogels have been applied in biomedical applications, most hydrogels show good adhesion only in dry environments, and adhesion can be significantly reduced or even lost in humid environment or underwater environment due to swelling and the formation of water molecular layer on the surface of hydrogels. Therefore, underwater high adhesion hydrogels have always been a hotspot and difficulty in this field. The surface modification on the macro and dynamic key water gel, the combination of regulation and control of dynamic water gel super molecular functional groups on the surface of the water gel, arrangement, so that it can rapidly with the substrate surface hydrophobic forces, thus successfully gives rapid and reversible hydrogels underwater strong adhesion, solved the difficult problem underwater adhesive. Especially in bone and cartilage tissue repair, in the context of fluid or blood containing potential applications.

## 5. Conclusions

Hydrogel is a very important material in bone and cartilage regeneration. It is rich in water and its physical and chemical properties can be changed in a wide range of ways. For a long time, researchers have conducted a lot of research on hydrogels, and the performance of hydrogels has been greatly enhanced, and the fields of application have also been expanding. 

However, several key difficulties of hydrogels have not been resolved, including: (1) the clinical medical application of hydrogels requires more rigorous testing. The US Food and Drug Administration (FDA) has only approved a few hydrogels for clinical applications; (2) the mechanical properties of hydrogels need to be enhanced to be applied to more fields; (3) the combination of hydrogel formulations and advanced biological manufacturing technology has great potential, but it still needs to be strictly optimized, so as to meet the needs of suitable biological manufacturing; (4) the printed hydrogel structure may be dynamically adjusted. When designing the material, the time dimension needs to be considered to form 4D printing. Nevertheless, the researchers believe that in the near future, with the continuous advancement of technology and methods, hydrogels with enhanced properties can be prepared. Eventually, engineering hydrogels will also be widely used according to their needs and optimal design.

## Figures and Tables

**Figure 1 materials-14-00235-f001:**
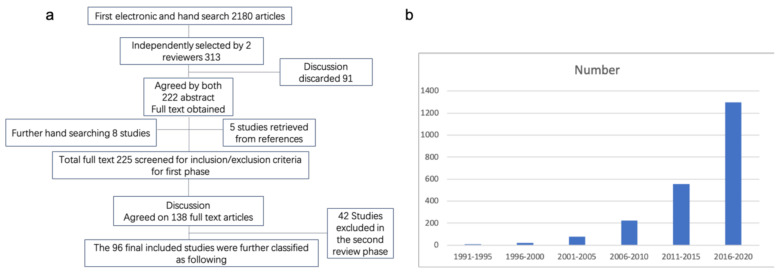
(**a**) Search strategy used to identify the included studies. (**b**) Graph representing the publication of hydrogel applicability shown with years.

**Figure 2 materials-14-00235-f002:**
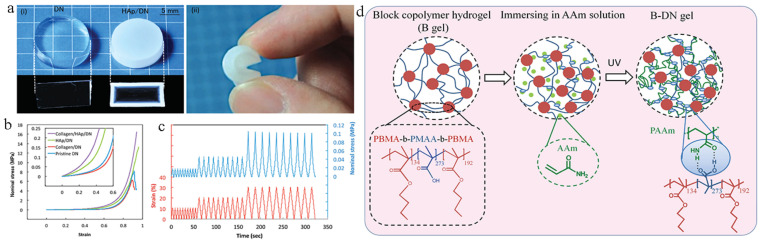
(**a**) Image of HAp/DN gel (hydroxyapatite nanospheres/dual network gel) (**i**) cross-sectional image; (**ii**) HAp/DN gel shows high elasticity. (**b**) Compressive stress-strain curves of HAp/DN, pristine DN, collagen/DN, and collagen/HAp/DN gels, and the inserted figure is the enlarged view of the low strain region. (**c**) Repeated compressive tests of HAp/DN for ten times at each strain; 10%, 20%, and 30% [22]. (**d**) Structure of block copolymer gel (B gel), the synthesis procedure of the double-network gel (B-DN gel), and the structure of B-DN gel with hydrogen bonds [25].

**Figure 3 materials-14-00235-f003:**
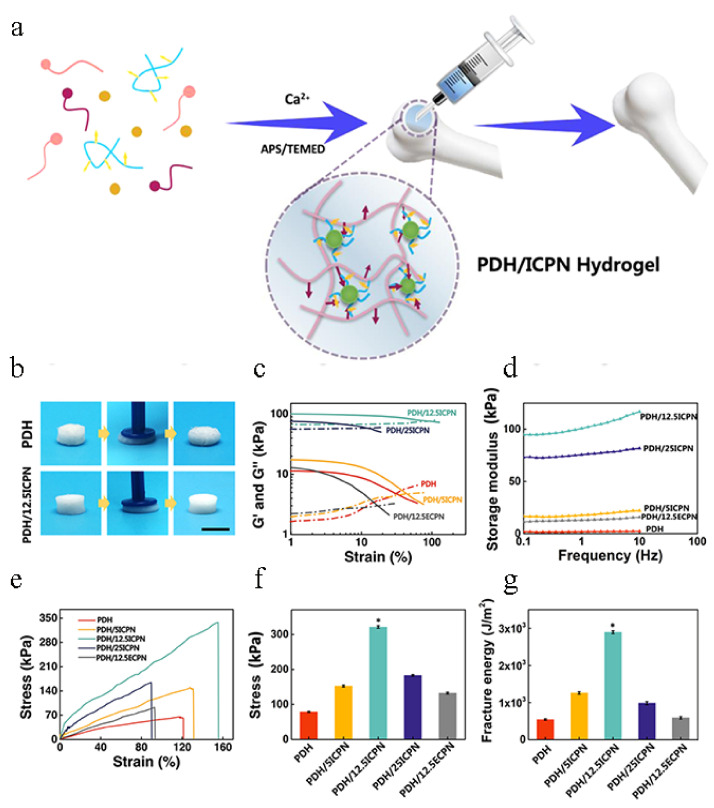
(**a**) Schematic illustration of the fabrication of the injectable PDH/ICPN hydrogel for bone regeneration. (**b**) Optical images of the PDH and PDH/12.5ICPN nanocomposite hydrogels under excessive compression (scar bar: 1 cm). (**c**) Strain-dependent (ω = 10 rad/s) (solid lines represent G′ and dashed line represent G″) and (**d**) frequency-dependent (at a strain of 1%) oscillatory rheological analysis results. (**e**) Stress-strain curves of the nanocomposite hydrogels with different calcium phosphate (CaP) contents. (**f**) Fracture stress and (**g**) fracture energy of PDH, PDH/mICPN, and PDH/12.5ECPN hydrogels (“*”: *p* < 0.05, indicating significant differences) [33].

**Figure 4 materials-14-00235-f004:**
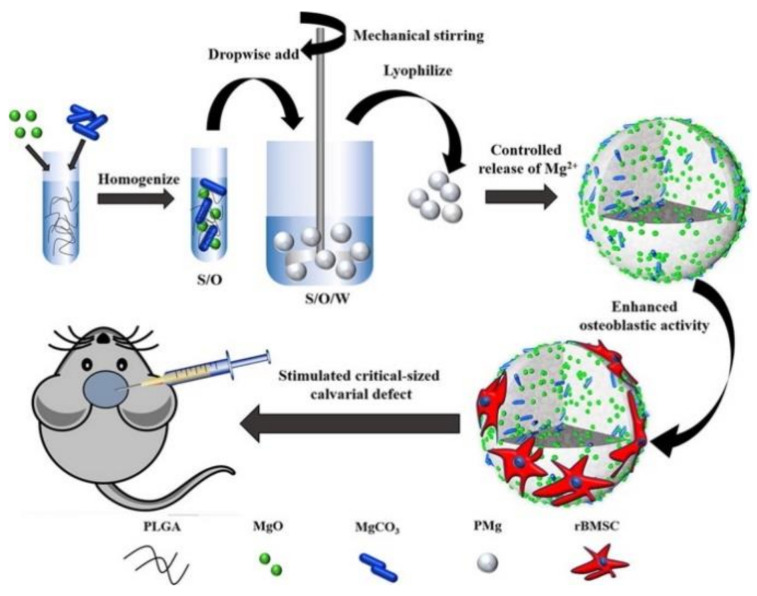
Using poly (lactic acid-glycolic acid) (PLGA), magnesium oxide (MgO), and magnesium carbonate (MgCO_3_) as raw materials, a series of PLGA\MgO\MgCO_3_ (PMg) composite microspheres were prepared by the emulsion method. Controlled release of Mg^2+^ [39].

**Figure 5 materials-14-00235-f005:**
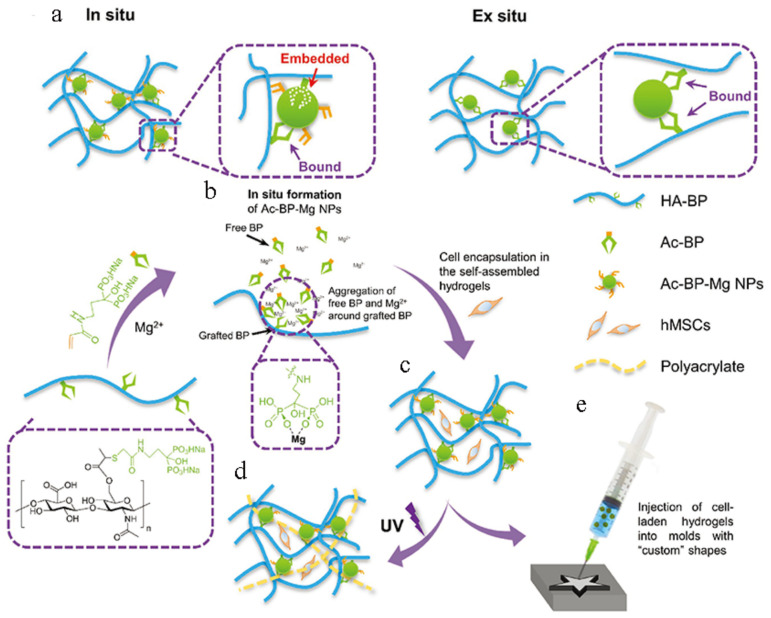
The in-situ formation of Ac-BP-Mg NPs enhances the mechanical properties of the HA-BP-Mg nanocomposite hydrogels. (**a**) In the in-situ hydrogels, some grafted bisphosphonates (BPs) of the HA-BP macromer are embedded into the in-situ-formed Ac-BP-Mg NPs (average particle size: 26.38 ± 6.46 nm), thereby strengthening the hydrogel network crosslinking, and the others are bound to the surface of NPs; for the ex situ hydrogels, mixing HA-BP with ex situ-formed MgSiO_3_ NPs (average particle size: 22.85 ± 4.04 nm) only leads to formation of weak surface interaction between the grafted BPs and the ex situ nanoparticles (NPs). Schematic illustration of the fabrication of the self-assembled HA-BP-Mg nanocomposite hydrogels. (**b**) The in-situ self-assembly of Ac-BP-Mg NPs around the grafted BP groups of the HA-BP macromer via BP-Mg^2+^ coordination. (**c**) The self-assembled cell-laden hydrogels are stabilized by the Ac-BP-Mg NPs. (**d**) UV-initiated radical polymerization of the acrylate groups at the surface of Ac-BP-Mg NPs leads to the secondary crosslinking and a further increase in the mechanical property of the self-assembled HA-BP-Mg nanocomposite hydrogels. (**e**) The cell-laden hydrogels can be injected and rapidly remolded to conform to the geometry of the injection sites [52].

**Figure 6 materials-14-00235-f006:**
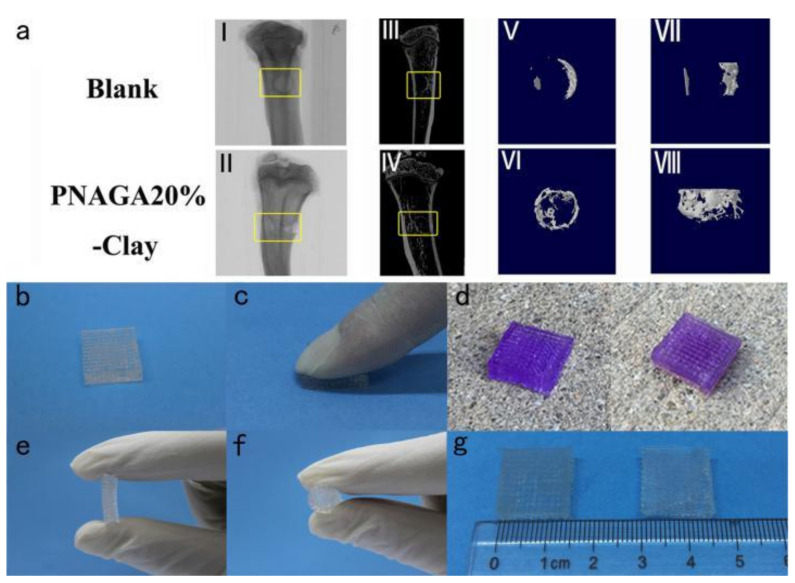
Characterization of implants and new bone formation by micro-CT (**a**(**I**,**II**)), micro-CT reconstruction images (**a**(**III**,**IV**)), and reconstructed 3D models (**a**(**V**–**VIII**)) of the new bones. Sequential fluorescent labeling blank photographs of PNAGA20%-Clay scaffolds showing their ability to resist finger compression (**b**,**c**), car wheel pressing (**d**), left and right denote before and after pressing), and hand folding (**e**,**f**). The scaffold is very stable even after immersing in water for a long time (**g**), left and right denote before and after water immersion for 3 months) [71].

**Figure 7 materials-14-00235-f007:**
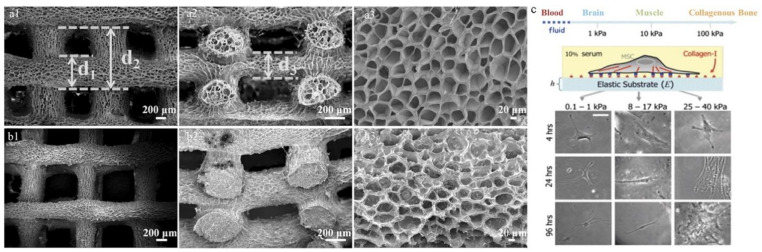
SEM images of the printed porous PNT-35%-6 hydrogel scaffolds (**a1**–**a3**) and the PNT-35%-6-β-TCP-22% hydrogel scaffolds (**b1**–**b3**). Top-down micrographs (**a1**,**b1**), cross-sectional view (**a2**,**b2**), and high magnification view of the surfaces (**a3**,**b3**), where *d*1 is the experimental printed filament diameter, *d*2 is the fiber spacing, and *d*3 is the layer thickness in the vertical dimension of each scaffold [77]. (**c**) The effect of material stiffness on cellular adhesion, differentiation behavior, and cell shape; the cells show branched and spindle shapes. The scale bar is 20 μm [76].

**Figure 8 materials-14-00235-f008:**
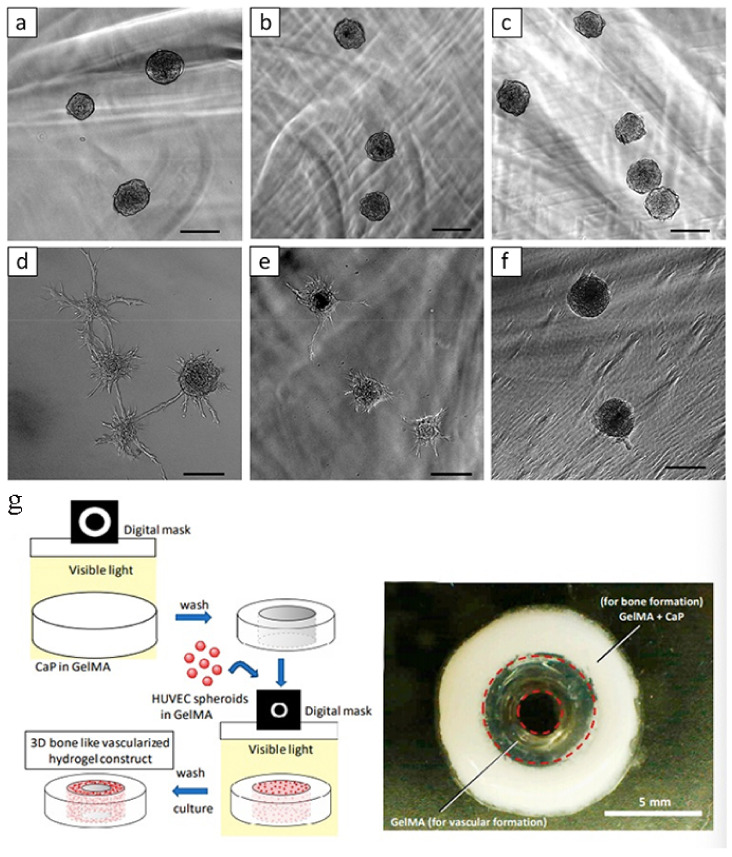
Three-dimensional in vitro angiogenesis of human umbilical vein epithelial cells (HUVEC) spheroids in the different concentrations of GelMA ((**a**,**d**) 5%, (**b**,**e**) 7.5%, and (**c**,**f**) 10%) just after polymerization of GelMA (**a**–**c**), and after 1 day of culture (**d**–**f**). Bars = 100 μm. (**g**) Schematic illustration of fabrication process for 3D hydrogel constructs. A photograph of 3D hydrogel constructs for vascular and bone formation. Bar = 5 mm [79].

**Figure 9 materials-14-00235-f009:**
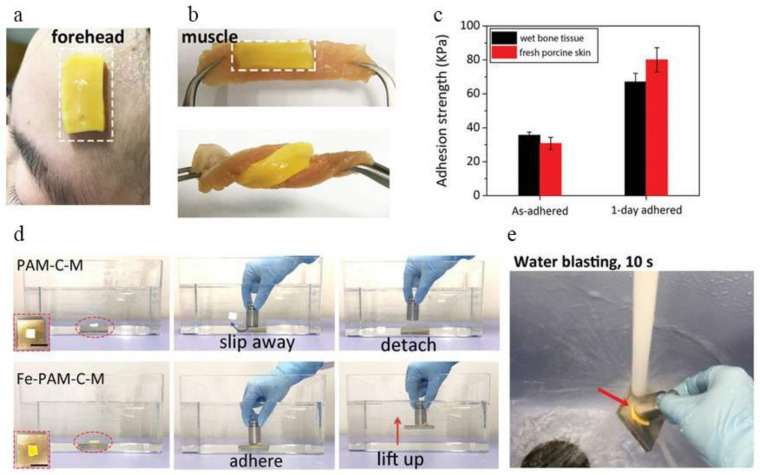
Wet tissue adhesiveness of Fe^3+^-polyacrylamide-hydrophobic stearyl methacrylate-N,N′-methylenebisacrylamide (Fe-PAM-C-M). (**a**) Photographs showing that the Fe-PAM-C-M hydrogel readily adhered to the forehead of one of the authors after perspiring. The humidity of the skin is higher than 120%. The size of the hydrogel applied on the skin is 10 mm in width, 20 mm in length, and 2 mm in thickness. (**b**) Photographs of a Fe-PAM-C-M hydrogel adhered to porcine muscle. No detachment or crack was observed between the hydrogel and tissue regardless of distortion. (**c**) The wet adhesion strength of Fe-PAM-C-M to porcine skin and bone tissues. The preload and contact time for tissue adhesion experiments are 5 kPa and 120 s. (**d**) Demonstration of underwater adhesion. The as-prepared hydrophilic PAM-C-M hydrogel was non-adhesive and slipped away from the metal block surface underwater, while the hydrophobic Fe-PAM-C-M hydrogel firmly adhered to the metal block surface and was able to lift the block (200 g) up underwater. (**e**) Photograph showing that the adhesion between the hydrogel and substrate is strong enough to resist water blasting for 10 s [95].

**Table 1 materials-14-00235-t001:** Mechanical properties and examples of several hydrogels [12].

Materials	Tensile Strength	Tensile Modulus	Compressive Strength	Compressive Modulus
Traditional single network hydrogel [15,16]	1–100 kPa	<100 kPa	10–100 kPa	1–100 kPa
Double network hydrogel [22]	10 MPa	1 MPa	60 MPa	100 kPa
Tetra-PEG hydrogel [33,34]	200 kPa	90 kPa	27 MPa	100 kPa
Topological [39]	20 kPa	-	-	350 kPa
Macromololecular microsphere composite hydrogel [29,45]	540 kPa	270 kPa	78.6 MPa	-
Nanocomposite hydrogel [46]	255 kPa	16 kPa	3.7 MPa	38 kPa

## Data Availability

Data sharing not applicable. No new data were created or analyzed in this study. Data sharing is not applicable to this article.

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
