# Peer review of "Applications of Hydrogel with Special Physical Properties in Bone and Cartilage Regeneration"

_materials, 2021, doi:10.3390/ma14010235_

Round 1

Reviewer 1 Report

In the manuscript entitled “Applications of hydrogels with special physical properties in bone and cartilage regeneration”, Lin and coworkers reviewed recent hydrogel research advances in the field of bone and cartilage regeneration. This review contains recent literatures, and current limitations of hydrogels in this field and future directions are summarized. I have some comments, mostly about editing, as below:

  1. Page 1, Line 34: It is mentioned that “They are composed of polysaccharides and proteins.” Do you mean “human tissues”? If so, this sentence should be connected to the previous one, otherwise it can mislead that all hydrogels are based on natural polymers.
  2. “Crosslink” with and without hyphen (“cross-link”) have been used throughout the manuscript. It should be unified.
  3. There are some abbreviations used without descriptions (e.g. DMOG, MPEG-PLA, GDF5, DDH, GAGNPs, Fe-PAM-C-M). Please add their descriptions.
  4. Page 5, line 191: About calcium phosphate nanoclusters, can they be considered as hydrogels? It seems like a highly viscous non-crosslinked material.
  5. Caption of Figure 3: “g)” Osteoporotic bone should be “d)”.
  6. Page 7, line 229: “The microsphere "realize"” should be changed.
  7. Page 9, line 301: Please edit the sentence “maintaining cell viability can also improve cell viability”
  8. Page 9, line 319: Please edit the sentence “And drug delivery systems.”.
  9. Page 9, line 325: “the hydrogel they developed…” Who are “they”?
  10. Page 11, line 395, “calcium phosphate” is repeated.
  11. This is a suggestion that the subtitle of 3.3.5 is bit unclear and it might be better to change. E.g. “underwater adhesive hydrogel”.
  12. In general, please have a thorough read and revise carefully.

Author Response

Response to Reviewer 1 Comments

In the manuscript entitled “Applications of hydrogels with special physical properties in bone and cartilage regeneration”, Lin and coworkers reviewed recent hydrogel research advances in the field of bone and cartilage regeneration. This review contains recent literatures, and current limitations of hydrogels in this field and future directions are summarized. I have some comments, mostly about editing, as below:

  • Page 1, Line 34: It is mentioned that “They are composed of polysaccharides and proteins.” Do you mean “human tissues”? If so, this sentence should be connected to the previous one, otherwise it can mislead that all hydrogels are based on natural polymers.

Response1: Thanks for the referee’s kind suggestion. "They" means human tissue. We change the sentence “At the same time, the physical and chemical properties of hydrogels are very similar to most human tissues. They are composed of polysaccharides and proteins.” to “At the same time, the physical and chemical properties of hydrogels are very similar to most human tissues which are composed of polysaccharides and proteins.” in our revised manuscript. (Page1, Line 34)

2、“Crosslink” with and without hyphen (“cross-link”) have been used throughout the manuscript. It should be unified.

Response 2: Thanks for the referee’s kind suggestion. I agree on the spelling of the word "crosslink" without hyphen in our revised manuscript.

3、There are some abbreviations used without descriptions (e.g. DMOG, MPEG-PLA, GDF5, DDH, GAGNPs, Fe-PAM-C-M). Please add their descriptions.

Response 3: Thanks for the referee’s kind suggestion. We have added the full name before each abbreviation in our revised manuscript.

GAGNPs: Glycosaminoglycan nanoparticles

DMOG: Dimethyloxallyl glycine

MPEG-PLA: Methoxy polyethylene glycol polylactic acid

Fe: Fe3+ ions

PAM: Polyacrylamide

C: hydrophobic stearyl methacrylate (C18)

M: N,N′‐methylenebisacrylamide

GDF5: Growth differentiation factor 5

DDH: Developmental dysplasia of hip

4、Page 5, line 191: About calcium phosphate nanoclusters, can they be considered as hydrogels? It seems like a highly viscous non-crosslinked material.

Response 4: Thanks for the referee’s kind suggestion. Calcium phosphate nanoclusters is not a highly viscous non-crosslinked material. So we have deleted the wrong example. And we have added the new example in our new manuscript. At the same time, we have changed the Figure 3 related with the previous example. And we've got the copyright agreement for the new Figure 3. (Page6, Line 206)

5、Caption of Figure 3: “g)” Osteoporotic bone should be “d)”.

Response 5: Thanks for the referee’s kind suggestion. We have deleted the previous Figure 3 and legend in our revised manuscript. So there is no “g)” now.

6、Page 7, line 229: “The microsphere "realize"” should be changed.

Response 6: Thanks for the referee’s kind suggestion. We have changed the sentence of  “The microspheres realize the controlled release of Mg2+ to “The release of Mg2+ can be controlled by the microsphere.” (Page7, Line 286)

7、Page 9, line 301: Please edit the sentence “maintaining cell viability can also improve cell viability”

Response 7: Thanks for the referee’s kind suggestion. We have changed the sentence of “Researchers found that hyaluronic acid-based hydrogels have better biocompatibility, and maintaining cell viability can also improve cell viability” to “Researchers found that hyaluronic acid-based hydrogels have better biocompatibility and can improve cell viability.”

(Page9, Line 364)

8、Page 9, line 319: Please edit the sentence “And drug delivery systems.”.

Response 8: Thanks for the referee’s kind suggestion. We deleted the words of “And drug delivery systems.” (Page9, Line 383)

9、Page 9, line 325: “the hydrogel they developed…” Who are “they”?

Response 9: Thanks for the referee’s kind suggestion. “They” means “zheng et al.” We have changed it in our revised manuscript. (Page9, Line 389)

10、Page 11, line 395, “calcium phosphate” is repeated.

Response 10: Thanks for the referee’s kind suggestion. We deleted the words of “calcium phosphate”. (Page11, Line 464)

11、This is a suggestion that the subtitle of 3.3.5 is bit unclear and it might be better to change. E.g. “underwater adhesive hydrogel”.

Response 11: Thanks for the referee’s kind suggestion. We changed the sentence of “The hydrogel remains viscous in humid environments” to “Underwater adhesive hydrogel”. (Page12, Line 498)

12、In general, please have a thorough read and revise carefully.

Response 12: Thanks for the referee’s kind suggestion. We have a thorough read and revise carefully.

Reviewer 2 Report

This review article gives an indepth insight to the application of hydrogel in bone and cartilage regeneration. Minor revision recommended.

1) It would be ideal if a graph representing the publication of hydrogel applicability is shown with years so that it gives readers an overview in a glance to the increase in demand of hydrogel.

2) Figure 5 is not clear, please make it more readable.

3) Can the incompatibility of the certain hydrogels be also included? Such as hydrogels based on polymethacrylate hydrogels

4) The author should also include photocurable hydrogels and its implementation in 3d printing.

5) The material characteristic features associated with hydrogels listed in the application category should be included.

6) Some of the references listed have missing page numbers such as ref 58, 92.

Author Response

Comments and Suggestions for Authors

Response to Reviewer 3 Comments

This review article gives an in depth insight to the application of hydrogel in bone and cartilage regeneration. Minor revision recommended.

1) It would be ideal if a graph representing the publication of hydrogel applicability is shown with years so that it gives readers an overview in a glance to the increase in demand of hydrogel.

Response 1: Thanks for the referee’s kind suggestion. We have make a graph representing the publication of hydrogel applicability is shown with years in our revised manuscript. (Figure 1b, Page2, Line 107)

2) Figure 5 is not clear, please make it more readable.

Response 2: Thanks for the referee’s kind suggestion. We have modified the Figure 5 to make it readable.(Page 8, Line 347)

3) Can the incompatibility of the certain hydrogels be also included? Such as hydrogels based on polymethacrylate hydrogels

Response 3: Thanks for the referee’s kind suggestion. We discussed this in the discussion section. The incompatibility of some hydrogels may also affect the application of hydrogels in bone and cartilage tissue engineering. Polymethacrylate hydrogels, which are generally non-toxic in their own right, cause inflammation and tissue reactions mainly due to soluble substances in the polymer, which generally include residual raw monomers, low molecular weight polymers, catalysts and other additives. Other hydrogels, which polymerize with ultraviolet light, can also cause cell death if the amount of photoinitiator is too high. So the issue of incompatibility needs to be carefully considered and the impact can be minimized by properly designing hydrogels. (Page 14, line 599-604)

4) The author should also include photocurable hydrogels and its implementation in 3d printing.

Response 4: Thanks for the referee’s kind suggestion. We have added the content about photocurable hydrogels and its implementation in 3D printing. (Page 10, Line 440-445)

5) The material characteristic features associated with hydrogels listed in the application category should be included.

Response 5: Thanks for the referee’s kind suggestion. We have added the content in our manuscript about the material characteristic features associated with hydrogels listed in the application category. (Page9-10, Line 418-429)

6) Some of the references listed have missing page numbers such as ref 58, 92.

Response 6: Thanks for the referee’s kind suggestion. We have added the page numbers in all the reference.

Reviewer 3 Report

Applications of hydrogel with special physical properties in bone and cartilage regeneration is very interesting paper.

Line 53: This article describes the morphology and mechanical properties of hydrogels used in bone and cartilage engineering.  (The morphology and mechanical properties are depending of the Synthesis method. Can you give an Information About the Synthesis method)

Line 61: the improved processing of Hydrogel. What is a processing, and what is an improvement.

Line 75: Is conclusion missing in your search strategy?

Line 84: 2.2.1. Inclusion criteria (why did you not inclue an origine of hydrogel in consideration)

Line 136: the mineralized hydroxyapatit nanosphere (which Mineral?). What is size of particles in Hydrogel?

Line 152: this gel requires a high ion concentration (which value: approx. g/L)

Line 187: What is an Advantage of nanaocomposite Hydrogels in comparison to other Hydrogels?

Author Response

Comments and Suggestions for Authors

Response to Reviewer 2 Comments

Applications of hydrogel with special physical properties in bone and cartilage regeneration is very interesting paper.

1、Line 53: This article describes the morphology and mechanical properties of hydrogels used in bone and cartilage engineering.  (The morphology and mechanical properties are depending of the Synthesis method. Can you give an Information About the Synthesis method)

Response 1: Thanks for the referee’s kind suggestion. The synthetic methods of hydrogels include physico-mechanical method, chemical method and physico-chemical combination method.

2、Line 61: the improved processing of Hydrogel. What is a processing, and what is an improvement.

Response 2: Thanks for the referee’s kind suggestion. From Line 404 to Line 538, several improvements in the processing methods of hydrogels for practical clinical application are introduced respectively.

  • The application of 3D printing to process hydrogels can make them better fit the shape of bone defects.
  • To repair cartilage and bone tissue defects at the same time, hydrogels are integrated processing.
  • To allow the hydrogels to retain their viscosity in a humid environment. it is required to combine macroscopic surface modification with hydrogels with dynamic bonds to control the arrangement of supramolecular functional groups in the dynamic hydrogels on the surface of the hydrogels, so that the hydrogels have an effect on biological tissues. Through this modification, the hydrogels retain their viscosity in a blood-filled environment.

3、Line 75: Is conclusion missing in your search strategy?

Response 3: Thanks for the referee’s kind suggestion. Yes, there is no conclusion, we have forgot to add the conclusion. So the sentence of “In the end, a total of 95 articles was included in the study.” was added in line 75.

  • Line 84: 2.2.1. Inclusion criteria (why did you not inclue an origine of hydrogel in consideration)

Response 4: Thanks for the referee’s kind suggestion. We have added the criteria of “an origine of hydrogel” in our revised manuscript. (Page3, Line 89)

5、Line 136: the mineralized hydroxyapatit nanosphere (which Mineral?). What is size of particles in Hydrogel?

Response 5: Thanks for the referee’s kind suggestion. The HAp nanocrystals with spiny spherical shape measuring ≈ 200-600 nm in diameter. And we have added the information in the sentence. (Page 4, Line 141)

6、Line 152: this gel requires a high ion concentration (which value: approx. g/L)

Response 6: Thanks for the referee’s kind suggestion. The ion concentration 2.0 × 10−3 M or higher.

7、Line 187: What is an Advantage of nanaocomposite Hydrogels in comparison to other Hydrogels?

Response 7: Thanks for the referee’s kind suggestion. Nanocomposite hydrogel generally improves the mechanical properties of hydrogels by adding NPs and introducing bioactive components, compared to other hydrogels. In addition, normal bone tissue consists of organic and inorganic components, and nanocomposite is added to hydrogels to mimic normal human tissue as much as possible.
